# Welding Characteristics of Laser-MIG Hybrid Welding of Arc-Welded Aluminum Profiles for High-Speed Trains

**DOI:** 10.3390/ma16010404

**Published:** 2023-01-01

**Authors:** Lingzhi Du, Zhibin Yang, Xing Wang

**Affiliations:** School of Materials Science and Engineering, Dalian Jiaotong University, Dalian 116028, China

**Keywords:** laser-MIG hybrid welding, arc-welded aluminum alloy profiles, weld formation, microstructure characteristics, mechanical properties

## Abstract

To expand the application of laser-MIG hybrid welding in the arc-welded aluminum alloy profile manufacturing for high-speed trains, the laser-MIG hybrid welding characteristics of 4 mm thick A6N01S-T5 arc-welded aluminum profiles were studied in this work. The welding parameters were optimized using the orthogonal test; the microstructure and properties of the joints were analyzed. The results showed that the optimal welding parameters were: welding speed 1.0 m/min, laser power 2.7 kW, arc current 200 A, spot diameter 0.8 mm, heat source distance 3 mm, and defocusing amount 0 mm. The columnar crystals and dense dendrites were observed near the fusion line and in the weld center, respectively, and the grains in the heat-affected zone were slightly coarse. The microhardness of the softening zone in the heat-affected zone was the lowest. The average tensile strength of the joints was 212 MPa, about 86% of the base metal. The samples fractured in the heat-affected zone, and the fractures showed typical plastic fracture characteristics. The results showed that laser-MIG hybrid welding has good applicability and feasibility for the arc-welded profile welding manufacturing.

## 1. Introduction

The light weight of high-speed trains has always been one of the goals of the rail transit industry. Aluminum alloy has been widely used in high-speed train body manufacturing due to its advantages of low density, high specific strength, good machinability, weldability, and excellent corrosion resistance. A6N01S aluminum alloy belongs to the Al-Mg-Si series of heat-treatable aluminum alloy, which is commonly used in the side wall and ceiling of high-speed trains and is the main material for high-speed train manufacturing [1,2,3].

At present, the welding of aluminum alloy profiles for high-speed trains is mainly based on MIG welding. There are many problems in the welding process, such as large heat input, low welding speed, and joint softening [4,5,6,7]. In view of the characteristics of its welding process, in order to achieve weld penetration, aluminum profiles generally need to preset arc-welding grooves with a small blunt edge height and large groove angle [8,9,10]. Laser-MIG hybrid welding uses a combination of laser and arc as a hybrid heat source, combines the advantages of laser beam welding and MIG welding, and has several undoubted superiorities: higher welding speed [11], deeper weld penetration [12], smaller heat input and narrower heat-affected zone [13], stronger assembly gap adaptability [14,15], less residual stress and deformation [16], and superior mechanical properties [17]. In 2003, Dilthey and Reich successfully used the laser-MIG hybrid welding method to weld an aluminum alloy bus compartment of ICE Trains. The total length of the weld was more than 1 km. The results showed that the quality of laser-MIG hybrid welding fully met the production requirements, and the welding deformation was greatly reduced compared with MIG welding. Li et al. [18] used the laser-MIG hybrid welding method to weld the 6N01 aluminum profile of a high-speed train side wall, and compared the results with the performance of MIG-welded joints. The results showed that the tensile strength of laser-MIG hybrid welded joints was about 10% higher than that of the MIG-welded joints, and the softening of joints was greatly improved. Jiang et al. [19] used the double-sided laser-MIG hybrid welding method to weld the 30 mm thick 5083 aluminum alloy. The results showed that the mechanical properties of the laser-MIG hybrid welded joints were better and the welding efficiency was higher. A large number of studies have shown that using laser-MIG hybrid welding technology to weld aluminum profiles has obvious technical and cost advantages [20,21,22,23]. At present, the groove form of the aluminum alloy profiles for high-speed trains is completely designed for the characteristics of MIG welding. In order to expand the application of laser-MIG hybrid welding technology in the welding of arc-welded aluminum alloy profiles, it is necessary to carry out the corresponding basic research.

In this paper, 4 mm thick A6N01S-T5 arc-welded aluminum profiles were welded by laser-MIG hybrid welding. The orthogonal test was used to optimize the welding parameters, and the microstructure and properties of the optimized joints were studied. The feasibility of laser-MIG hybrid welding in the welding of arc-welded profiles of high-speed train aluminum alloy car bodies was discussed.

## 2. Materials and Methods

### 2.1. Materials

The base material is the hollow arc-welded profile of A6N01S-T5 aluminium alloy. The size of the test plate is 500 mm × 200 mm, the thickness of the welding part is 4 mm, and the groove form is shown in Figure 1. The filler material is ER5356 aluminum alloy filler wire with a diameter of 1.2 mm. The chemical composition of base metal and welding wire is shown in Table 1. Before welding, the surface of the sample was mechanically polished and wiped with acetone to remove the oxide film and oil stain.

### 2.2. Methods

The laser-MIG hybrid welding experiments were carried out by a TRUMPF TruDisk 16003 disc laser in combination with a FRONIUS TPS 5000 CMT welding machine. During the welding process, the laser was in the front and the arc was in the back. The angles between the laser beam, welding gun, and testing plate were 80° and 60°, respectively. The shielding gas was 99.999% high-purity argon and the flow rate was 50 L/min. The schematic diagram of the laser-MIG hybrid welding method is shown in Figure 2.

Based on the previous process exploration, the optimization of welding parameters was carried out by orthogonal test with three factors: laser power, arc current, and defocusing amount. The welding speed, spot diameter, and heat source spacing was 1.0 m/min, 0.8 mm, and 3.0 mm, respectively. The orthogonal test is a three-factor three-level design scheme, as shown in Table 2: A represents laser power, B represents welding current, and C represents defocusing amount.

Metallography samples were taken from the weld cross-section, mounted in resin, ground by abrasive papers with different grits, then polished with polishing agents, and finally etched using Keller’s reagent for about 10 s. According to GB/T 26955-2011, the weld formation and microstructure were observed by KEYENCE VHX-1000E three-dimensional video microscope. With reference to GB/T 3323.1-2019, we used a XXG-2005 industrial X-ray flaw detector to detect weld internal porosity defects, with detection length 100 mm. Referring to GB/T 2654-2008, the microhardness distribution of the joint was measured by FM-700 microhardness tester. Referring to GB/T 2651-2008, the tensile properties of joints were tested by WDW-300E electronic universal testing machine. Referring to GB/T 223-2010, we used a WDW-300E electronic universal testing machine to perform joint bending test. The fracture morphology of tensile specimen was observed by ZEISS SUPRA55 scanning electron microscope. The Minitab 17 Statistical Software is used for the optimization of the results.

## 3. Results

### 3.1. Parameter Optimization

The weld-surface forming, cross-section formation, and porosity defects with different welding parameters were tested by an orthogonal test. The average tensile strength of the joints was used as the evaluation index of welding parameters. Based on the principle of range analysis, the range analysis results obtained were shown in Table 3. The weld surface, weld cross-section, and porosity were as shown in Table 3 and Figure 3. The results showed that the weld surface of the joints with various process parameters were all fine-formed, the penetrations were good, and there were no obvious defects. In addition, in the 2# and 4# joints, there existed some small-sized porosity defects; the remaining joints had no porosity defects. The maximum average tensile strength of the joint reaches 199 MPa, which was about 81% of the tensile strength of the base metal. *K* and *R* respectively represented range and variance of the tensile strength. Based on the magnitude of the *K* and *R*, the optimal value and priority order of the influencing factors could be confirmed. The greater the values of the *K* and *R*, the greater the influence of the factor on the tensile strength. Therefore, the order of the influence of the three factors on the average tensile strength of the joint was arc current > laser power > defocusing amount. The optimal combination of each factor was A3B1C1, and the optimization level of each factor was laser power 2.7 kW, arc current 200 A, and defocusing amount 0 mm.

The surface forming, cross-section formation, and porosity defect results of the joints with optimized welding parameters are shown in Figure 4. The test results showed that the weld had good formation without obvious defects, and had no obvious porosity defects in the weld.

### 3.2. Microstructure Characteristics

The microstructure characteristics of the joint are shown in Figure 4, including three regions: weld zone (WM), heat-affected zone (HAZ), and base metal (BM). There were columnar crystals near the fusion line of the arc-affected zone and the laser-affected zone, as shown in Figure 5a,b. The main reason was that the cooling rate in this area was fast and the temperature gradient was large. The liquid metal was preferentially grown in the molten pool in the opposite direction of the maximum heat dissipation direction of the boundary, and the growth of other-oriented grains was inhibited. The grains in the heat-affected zone were slightly coarse, which was mainly caused by the high welding heat input. The weld center of the arc-affected zone and the laser-affected zone was a typical as-cast dendritic structure, as shown in Figure 5c,d. The weld center of the arc-affected zone had a higher heat input and a slower cooling rate of the molten pool, and its grain size was larger than that of the laser-affected zone.

### 3.3. Mechanical Properties

The microhardness distribution of the joint is shown in Figure 6. The results showed that there was an obvious softening phenomenon in the welded joint. The microhardness value was the highest in the base metal; the microhardness values of the weld seam and the heat-affected zone decreased in different degrees. The adopted filler wire microhardness lower than the base metal and the alloying elements burned and lost during the welding process were the two main reasons for the microhardness’ decrease in the weld seam [24]. The lowest microhardness value was located in the heat-affected zone, and the microhardness value was about 57 HV, about 5 mm from the fusion line. At the same time, it was found that the microhardness value decreased to the lowest value in the range of 4–7 mm from the fusion line. The width of the softening zone was about 2–3 mm, which was mainly caused by the decrease of the dissolution of the strengthening phase β″ [25,26]. The microhardness results showed that the softening zone of the heat-affected zone was the weakest position of the joint.

The tensile testing results of the joints are shown in Table 4. The average tensile strength of the joint was about 212 MPa, which was about 86% of the base metal. The fracture position of the samples was in the softening zone of the heat-affected zone, mainly because of the significant softening phenomenon of the joint in this area. The microhardness test results of the joint were consistent with it. This area was the weakest link of the joint. The microstructure of the tensile specimen fracture is shown in Figure 7. A small number of tearing characteristics could be found in the macroscopic fracture, as shown in Figure 7a; the microscopic fracture showed a large number of dimples, and the dimples were deep and similar in size, as shown in Figure 7b. The fracture morphology showed typical plastic fracture characteristics.

The bending tests were carried out with the bending angle of 180° and indenter diameter of 20 mm, and the compressing velocity was 1 mm/min. The bending test results of the joints are shown in Table 5. The results showed that: after the 180° face-bending and root-bending test, the tensile surface morphology was fine, the surface was smooth, and no obvious surface crack was found. The results indicated that the bending properties of the laser-MIG hybrid welded joints were good and met the requirements.

## 4. Conclusions

The optimized laser-MIG hybrid welding parameters of a 4 mm thick arc-welded aluminum alloy profile were: welding speed 1.0 m/min, laser power 2.7 kW, arc current 200 A, spot diameter 0.8 mm, heat source spacing 3 mm, and defocus amount 0 mm. Based on the orthogonal test, the order of influence on the tensile strength of the joint was arc current > laser power > defocus amount.The microstructure of the joint weld center was a typical dendrite structure, and the grain size of the arc-affected zone was larger than that of the laser-affected zone; the columnar crystal structure was near the fusion line, and the grains in the heat-affected zone were slightly coarse. The heat-affected zone had an obvious softening phenomenon, which was the lowest hardness area of the joint.The average tensile strength of the joint reached up to 212 MPa, which was about 86% of the base metal. The sample fractured in the softening zone of the heat-affected zone, and the fracture showed typical plastic fracture characteristics. Bending specimens by 180° root bend and face bend results in a stretching, smooth, crack-free surface.The test results show that laser-MIG hybrid welding had good applicability and feasibility in high-speed train arc-welded aluminum alloy profile manufacturing.

## Figures and Tables

**Figure 1 materials-16-00404-f001:**
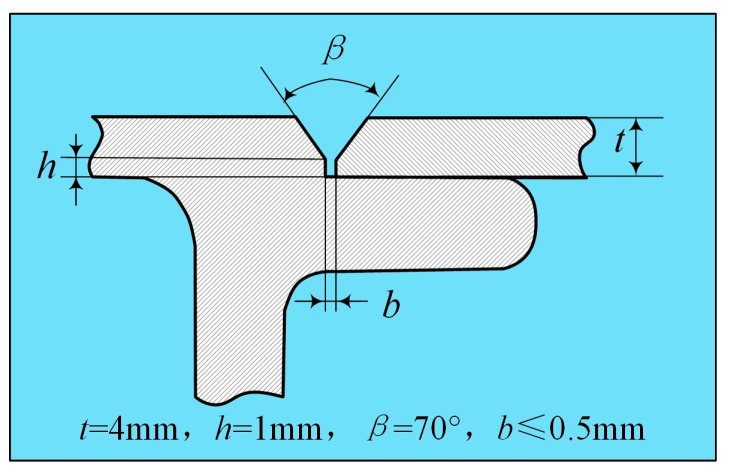
Schematic diagram of groove type for arc-welded aluminum alloy profiles.

**Figure 2 materials-16-00404-f002:**
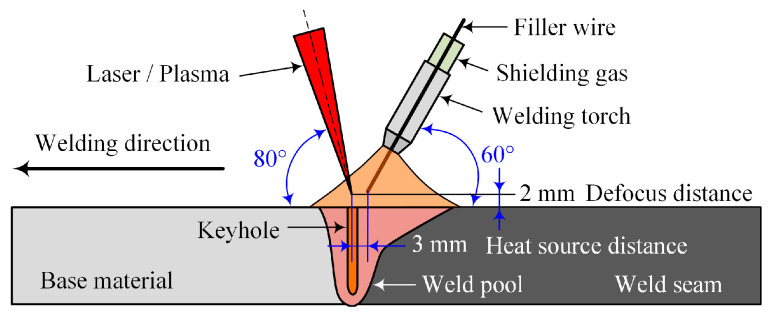
Schematic diagram of laser-MIG hybrid welding.

**Figure 3 materials-16-00404-f003:**
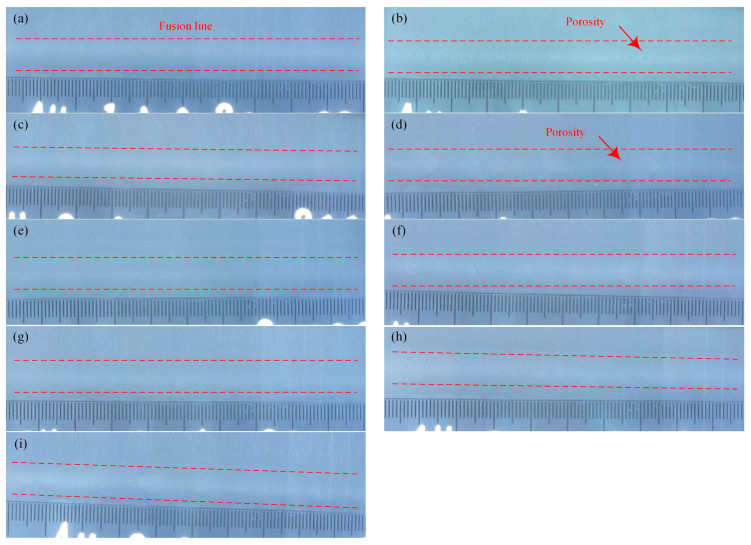
Porosity in the different weld seams of Table 3: (**a**) 1#; (**b**) 2#; (**c**) 3#; (**d**) 4#; (**e**) 5#; (**f**) 6#; (**g**) 7#; (**h**) 8#; and (**i**) 9#.

**Figure 4 materials-16-00404-f004:**
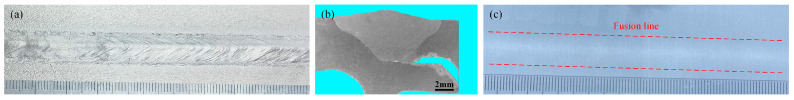
Weld quality of the optimized welding parameters: (**a**) surface; (**b**) cross-section; and (**c**) porosity.

**Figure 5 materials-16-00404-f005:**
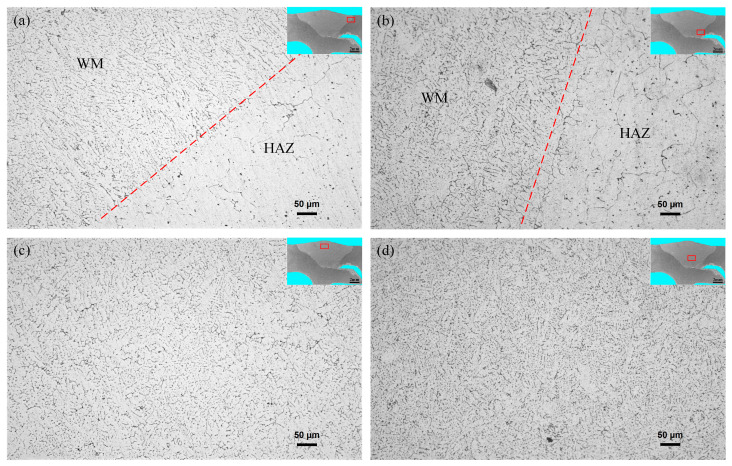
Microstructure of the joints: (**a**) near the fusion line of the arc-affected zone; (**b**) near the fusion line of the laser-affected zone; (**c**) weld center of the arc-affected zone; and (**d**) weld center of the laser-affected zone.

**Figure 6 materials-16-00404-f006:**
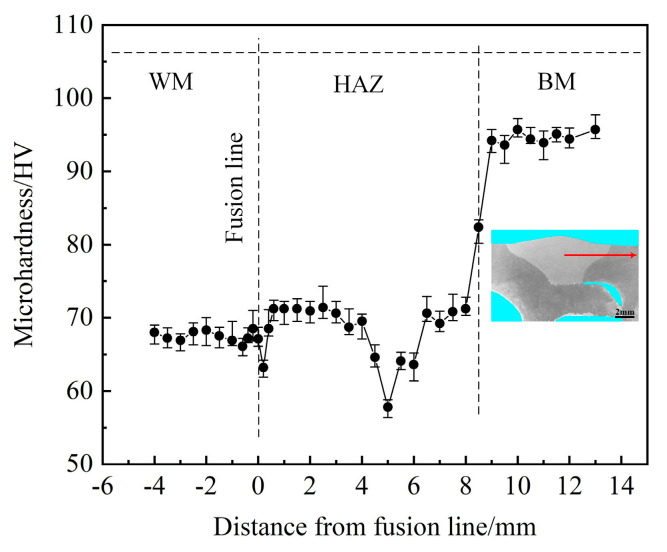
Microhardness distribution of the joint. The arrow represents the location and direction of the microhardness test.

**Figure 7 materials-16-00404-f007:**
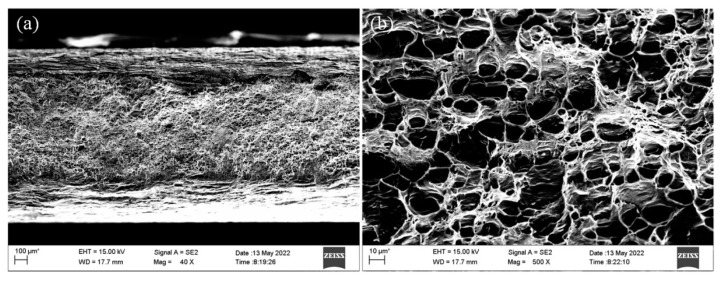
Fracture characteristics of the tensile specimen: (**a**) macroscopic feature; and (**b**) microscopic feature.

**Table 1 materials-16-00404-t001:** Chemical composition of the base material and filler wire (wt.%).

Material	Si	Fe	Cu	Mn	Mg	Cr	Zn	Ti	Al
A6N01S-T5	0.4~0.9	≤0.35	≤0.35	≤0.3	0.4~0.8	≤0.3	≤0.25	≤0.35	Bal.
ER5356	0.13	0.12	0.01	0.06	4.90	0.07	0.12	0.11	Bal.

**Table 2 materials-16-00404-t002:** Factor level scheme of process parameters was used in orthogonal experiment.

Level	Factor		
	A/kW	B/A	C/mm
1	2.5	200	0
2	2.3	190	−2
3	2.7	210	+2

**Table 3 materials-16-00404-t003:** Welding parameters, experimental, and range analysis results of the orthogonal test.

No.	Laser Power /kW	Arc Current /A	Defocusing Amount /mm	Surface	Cross-Section	Tensile Strength /MPa
1#	2.5	200	0	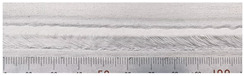	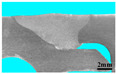	196
2#	2.5	190	−2	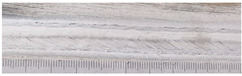	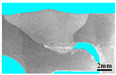	165
3#	2.5	210	2	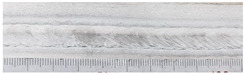	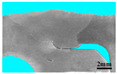	173
4#	2.3	200	−2	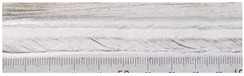	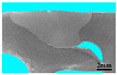	195
5#	2.3	190	2	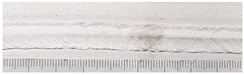	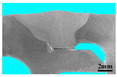	172
6#	2.3	210	0	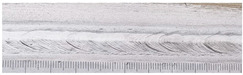	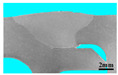	182
7#	2.7	200	2	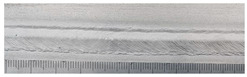	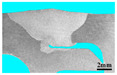	199
8#	2.7	190	0	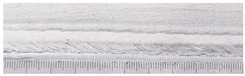	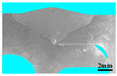	182
9#	2.7	210	−2	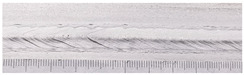	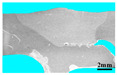	193
*K* _1_	177.7	196.7	186.3			
*K* _2_	182.8	172.8	184.2			
*K* _3_	191.4	182.4	181.4			
*R*	13.6	23.8	4.8			
Priorities	B > A > C					
Optimization	A3	B1	C1			

**Table 4 materials-16-00404-t004:** The tensile testing results of the joints.

No.	Tensile Strength/MPa		Joint Efficiency/%	Fracture Location	
	Sample	Average			
LS-1#	217	212	86	HAZ	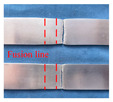
LS-2#	207				

**Table 5 materials-16-00404-t005:** The result of the bending tests.

No.	Sample Size/mm	Bending Angle 180°, Indenter Diameter 20 mm	Note
WQ-1#	300 × 25 × 3	no crack	Face bend	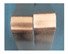
WQ-2#	300 × 25 × 3	no crack
WQ-3#	300 × 25 × 3	no crack	Root bend	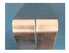
WQ-4#	300 × 25 × 3	no crack

## Data Availability

Not applicable.

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
