# Peer review of "Welding Characteristics of Laser-MIG Hybrid Welding of Arc-Welded Aluminum Profiles for High-Speed Trains"

_materials, 2023, doi:10.3390/ma16010404_

Round 1

Reviewer 1 Report

The authors present a straightforward study of the effect of three parameters on hybrid laser arc welding of aluminum. The study is well designed and executed; the conclusions are sound. Some improvement to the description of the methods and presentation of the results is needed as follows:

-How are the metallography samples prepared? Is an etchant used? Why is the micrograph tinted blue?

-More details on the range analysis would aid the reader in understanding how the parameters were optimized.

-How many Vickers indents were performed at each location. Please include error bars in Figure 5.

-The resolution of the images in table 3 are too low to distinguish differences between each processing condition. In particular, the labeled porosity for #2 is not readily apparent from the image. It is suggested to place the images in separate high resolution figure.

Author Response

Dear reviewer, Thank you for your comments concerning our submitted manuscript. Those comments are all valuable and very helpful for revising and improving the quality of the paper. We have studied comments carefully and have made corrections which we hope meet with approval. The main corrections in the revised manuscript are marked in red, and the point to point responds to the reviewer’s comments are as follows:

Comment 1: How are the metallography samples prepared? Is an etchant used? Why is the micrograph tinted blue?

Responses: We have added the preparation method of the metallography samples in the revised manuscript. And the keller’s reagent used as etchant in this work, and also added in a suitable place of the revised manuscript (in page 3, line 90-92). We have replaced the tinted blue micrographs with the normal micrographs (in page 6, line 149-151).

Comment 2: More details on the range analysis would aid the reader in understanding how the parameters were optimized.

Responses: According to the suggestion of the Reviewer, in order to aid the readers in understanding how the parameters were optimized, we have given some more details on the range and variance analysis of the tensile testing results: K and R respectively represented range and variance of the tensile strength. Based on the magnitude of the K and R, the optimal value and priority order of the influencing factors could be confirmed. The greater the values of the K and R, the greater the influence of the factor on the tensile strength (in page 3, line 114-117).

Comment 3: How many Vickers indents were performed at each location. Please include error bars in Figure 5.

Responses: There are 3 Vickers indents were performed nearby every location. Thank you for your suggestion, we have added the error bars in Fig. 5 in the revised manuscript (in page 7, line 166-167). This will make our trial data more rigorous.

Comment 4: The resolution of the images in table 3 are too low to distinguish differences between each processing condition. In particular, the labeled porosity for #2 is not readily apparent from the image. It is suggested to place the images in separate high resolution figure.

Responses: Considering the Reviewer’s suggestion, in order to improve the resolution of the images, we have changed the arrangement of the table 3 (in page 4, line 123). Meanwhile, we have placed the porosity images in separate high resolution figure 3 (in page 5, line 124-129), and we have also given the schematic of the fusion line of the weld seam of the X-ray photos. We have tried our best to improve the manuscript and made some changes in the revised manuscript. And here we did not list all of the changes but marked in red in revised submissions. We appreciate for your warm work earnestly, and hope that the corrections will meet with approval. Once again, thanks very much for your comments and suggestions.

Reviewer 2 Report

The submitted article is of poor quality, and not acceptable in its current form. Enough reservations are available that are listed below, can be reconsidered if addressed properly and correctly;

[1]. The technical essence of the utilization of hybrid LASER and MIG welding is still missing in the introduction section.

[2]. The references that provoke the utilization or pave the way towards this work are missing in the introduction, need to add valuable references.

[3]. I still doubt for the 4mm plate of Al-Mg-Si was not handled with a single TIG or MIG Source, need to add solid justification for moving toward the hybrid.

[4]. Why not opt for 2 or 3-mm wire of 5356 filler wire?

[5]. GB / T 26955-2011, GB / T 2654-2008, GB / 92 T 2651-2008, GB / T 223-2010, what are these standards, and from which society, Like ASTM?

[6]. Which software is used for the optimization of results?

[7]. On what basis you have concluded the optimized results?

[8]. What are K and R in Table 3?

[9]. The surface pictures in Table 3 are not clear.

[10].The term optimization throughout the article is completely vague, I didn’t find the right techniques for this.
How the authors have extracted the weld centers for arc and laser-affected zones in Fig 4(c, d)?

[11].  The justification for the microhardness in Fig. 5 is not enough.

[12]. Section 3.3 is totally inappropriate and not justifiable with the quality of the journal.

[13].  Where is the technical description of the bend test?

[14].References are directional with the specific country and need to revise appropriately.

Author Response

Dear reviewer, Thank you for your comments concerning our submitted manuscript. Those comments are all valuable and very helpful for revising and improving the quality of the paper. We have studied comments carefully and have made corrections which we hope meet with approval. The main corrections in the revised manuscript are marked in red, and the point to point responds to the reviewer’s comments are as follows:

Comment 1: The technical essence of the utilization of hybrid LASER and MIG welding is still missing in the introduction section.

Responses: Thanks for the reviewer’s suggestions, in the introduction section of the revised manuscript, we have added some detailed introductions of the Laser-MIG hybrid welding, especially of its main technical advantages (in page 1, line 35-39).

Comment 2: The references that provoke the utilization or pave the way towards this work are missing in the introduction, need to add valuable references.

Responses: The reviewer’s advice is very meaningful, and the relevant references are also the materials we desperately needed. However, after a long time of searching, we still did not find very valuable relevant literatures for this study. The laser-MIG hybrid welding is one of the common methods for joining the aluminum alloy, but we have not seen laser-MIG hybrid welding of arc-welded aluminum alloy profiles for high-speed trains. Therefore, the research purpose of this paper is to expand the application of laser-MIG hybrid welding in the arc-welded aluminum alloy profiles manufacturing for high-speed trains. And therefore, in the introduction section, we only can introduce several references on laser-MIG hybrid welding for the conventional constructions, without for the arc-welded profiles. We will continue to search relevant literatures in the follow-up research.

Comment 3: I still doubt for the 4mm plate of Al-Mg-Si was not handled with a single TIG or MIG Source, need to add solid justification for moving toward the hybrid.

Responses: Just as the reviewer’s said, the 4mm plate of Al-Mg-Si could be handled with a single TIG or MIG source. However, their penetration abilities are weak, and weld defect of incomplete fusion is easily appeared. Meanwhile, the TIG welding additional needs filler wire to obtain the suitable welds, which is rarely used for welding 4mm thick aluminum alloys. At present, the MIG welding is a very common joining method for aluminum alloys. But, with the increase of train speed, the mechanical properties of the traditional MIG welded joints could not meet the requirements, as well as the forming quality (especially of the deformation) reduces the beauty of the train body. The existing research results show that, compared with the MIG welding, the laser-MIG hybrid welding has higher welding speed (over 2 times increased), smaller residual stress and deformation (can avoid adjustment and repair after welding) and greater mechanical properties (tensile strength can increase more than 20%, fatigue limit can increase more than 15%), as described in the introduction section of the revised manuscript (in page 1, line 37-39, and page 2, line 46-50). The above descriptions are also some reasons for choosing the laser-MIG hybrid welding for the 4mm thick arc-welded aluminum alloy profiles in this work.

Comment 4: Why not opt for 2 or 3-mm wire of 5356 filler wire?

Responses: The diameter specifications of the filler wire for MIG welding are 0.8mm, 1.0mm, 1.2mm and 1.6mm. And the diameter of 1.2 mm is the commonest filler wire. Smaller diameter will result in the lower welding speed to obtain well weld seams, and larger diameter will result in the welding current increased and increase the welding deformation. Therefore, the filler wire of 1.2 mm in diameter was adopted in this work.

Comment 5: GB / T 26955-2011, GB / T 2654-2008, GB / 92 T 2651-2008, GB / T 223-2010, what are these standards, and from which society, Like ASTM?

Responses: The above standards are all from national standard of China. Their relevant regulations are similar to ISO or ASTM.

Comment 6: Which software is used for the optimization of results?

Responses: In this work, the Minitab 17 Statistical Software is used for the optimization of the results. Meanwhile, we have added the above introduction in section 2.2 of the revised manuscript (in page 3, line 101-102).

Comment 7: On what basis you have concluded the optimized results?

Responses: In this work, the tensile strength of the joints is the key evaluation indicator of the welding parameters (as described in page 3, line 106-107). Therefore, in this paper, through an orthogonal test analyzed the factors influencing the tensile strength, and finally concluded the optimized results.

Comment 8: What are K and R in Table 3?

Responses: In table 3, the K and R respectively represented range and variance of the tensile strength. Based on the magnitude of the K and R, the optimal value and priority order of the influencing factors could be confirmed. The greater the values of the K and R, the greater the in-fluence of the factor on the tensile strength. We have added the above descriptions in section 3.1 of the revised manuscript (in page 3, line 114-117).

Comment 9: The surface pictures in Table 3 are not clear.

Responses: Considering the Reviewer’s comment, in order to improve the resolution of the images, we have changed the arrangement of the table 3 (in page 4, line 123). Meanwhile, we have placed the porosity images in separate high-resolution figure 3 (in page 5, line 124-129), and we have also given the schematic of the fusion line of the weld seam of the X-ray photos.

Comment 10: The term optimization throughout the article is completely vague, I didn’t find the right techniques for this. How the authors have extracted the weld centers for arc and laser-affected zones in Fig 4(c, d)?

Responses: We are sorry for our confusions in this description. In order to clearly identify the locations of the weld center of the arc affected zone and laser affected zone, we have added the schematic diagrams in the microstructure pictures, as shown in Fig 5 (c) and (d) (in page 6, line 149-151).

Comment 11: The justification for the microhardness in Fig. 5 is not enough.

Responses: Considering the reviewer’s suggestion, we have added some detailed descriptions of the microhardness distributions of the joint (in page 6, line 154-158). Meanwhile, some references are added for giving explanations (in page 6, line 159, 163 and 164).

Comment 12: Section 3.3 is totally inappropriate and not justifiable with the quality of the journal.

Responses: Considering the review’s comment, in the parts of the microhardness distribution and bending property of the revised manuscript, we have added some descriptions and explanations (in page 6, line 154-159, page 7, line 182-183, and page 8, line 186-187). We hope this version could appropriate and justifiable with the quality of this journal.

Comment 13: Where is the technical description of the bend test?

Responses: The bending test was carried out according to GB / T 223-2010, as described in section 2.2 (in page 3, line 98-99). And the detailed technical descriptions are added in section 3.3 (in page 7, line 182-183): The bending tests were carried out with the bending angle of 180 ° and indenter diameter of 20 mm, and the compressing velocity was 1 mm/min.

Comment 14: References are directional with the specific country and need to revise appropriately.

Responses: In order to give some more explanations, we have added and deleted some references in the correct locations of the revised manuscript. And the authors of the cited references are come from China, Italy, UK, USA, Turkey, India, Portugal, Japan, Norway, Finland and so on. Meanwhile, there are no more than 2 papers by the same author. Therefore, in the revised manuscript, the references are no more directional with the specific country and author. We have tried our best to improve the manuscript and made some changes in the revised manuscript. And here we did not list all of the changes but marked in red in revised submissions. We appreciate for your warm work earnestly, and hope that the corrections will meet with approval. Once again, thank you very much for your comments and suggestions.
